## Research Article

**Key words:**
Membrane protein design; protein structural predictions; QTY code; water-soluble membrane transporters

**Author for correspondence:**
*Shuguang Zhang,
E-mail: Shuguang@mit.edu

CAMBRIDGE
UNIVERSITY PRESS

# Comparing 2 crystal structures and 12 AlphaFold2-predicted human membrane glucose transporters and their water-soluble glutamine, threonine and tyrosine variants

Eva Smorodina[1] , Fei Tao[2] , Rui Qing[2] , David Jin[3] , Steve Yang[4] and Shuguang Zhang[5]* 

[1]Laboratory for Computational and Systems Immunology, Department of Immunology, University of Oslo, Oslo, Norway; [2]Laboratory of Food Microbial Technology, State Key Laboratory of Microbial Metabolism, School of Life Sciences and Biotechnology, Shanghai Jiaotong University, Shanghai 200240, China; [3]Avalon GloboCare Corp., Freehold, NJ 07728, USA; [4]PT Metiska Farma, Daerah Khusus Ibukota, Jakarta 12220, Indonesia and [5]Laboratory of Molecular Architecture, Media Lab, Massachusetts Institute of Technology, 77 Massachusetts Avenue, Cambridge, MA 02139, USA

## Abstract

Membrane transporters including glucose transporters (GLUTs) are involved in cellular energy supplies, cell metabolism and other vital biological activities. They have also been implicated in cancer proliferation and metastasis, thus they represent an important target in combatting cancer. However, membrane transporters are very difficult to study due to their multispan transmembrane properties. The new computational tool, AlphaFold2, offers highly accurate predictions of three-dimensional protein structures. The glutamine, threonine and tyrosine (QTY) code provides a systematic method of rendering hydrophobic sequences into hydrophilic ones. Here, we present computational studies of native integral membrane GLUTs with 12 transmembrane helical segments determined by X-ray crystallography and CryoEM, comparing the AlphaFold2-predicted native structure to their water-soluble QTY variants predicted by AlphaFold2. In the native structures of the transmembrane helices, there are hydrophobic amino acids leucine (L), isoleucine (I), valine (V) and phenylalanine (F). Applying the QTY code, these hydrophobic amino acids are systematically replaced by hydrophilic amino acids, glutamine (Q), threonine (T) and tyrosine (Y) rendering them water-soluble. We present the superposed structures of native GLUTs and their water-soluble QTY variants. The superposed structures show remarkable similar residue mean square distance values between 0.47 and 3.6 Å (most about 1–2 Å) despite >44% transmembrane amino acid differences. We also show the differences of hydrophobicity patches between the native membrane transporters and their QTY variants. We explain the rationale why the membrane protein QTY variants become water-soluble. Our study provides insight into the differences between the hydrophobic helices and hydrophilic helices, and offers confirmation of the QTY method for studying multispan transmembrane proteins and other aggregated proteins through their water-soluble variants.

## Introduction

The common hallmark of almost all cancers and tumours is rapid, uncontrolled growth, cellular proliferation and metastasis (Yamamoto *et al.,* 1990; Macheda *et al.,* 2005; Airley and Mobasheri, 2007; Szablewski, 2013; Wang *et al.,* 2015; Barron *et al.,* 2016; Ancey *et al.,* 2018, Wu *et al.,* 2021). Such metastatic growth demands a constant supply of nutrients, especially sugars, particularly glucose and fructose and most cancer cells upregulate their glucose transporters (GLUTs) without obvious mutations (Godoy *et al.,* 2006; Barron *et al.,* 2016). It has been shown that glucose transporters GLUT1–14, especially GLUT 1–9, 11–12, 14 are involved in cancer metabolism and fuel cancer growth (Barron *et al.,* 2016; Ancey *et al.,* 2018). If these sugar transporters can be effectively targeted and specifically inhibited, the proliferation and metastasis of cancer cells may also likely be inhibited. Structural studies of these GLUTs may ultimately lead to significant clinical benefits for cancer patients.

GLUTs have different effects and involvement in various cancers (Table 1; Chandler *et al.,* 2003; Barron *et al.,* 2016). For example, overexpression or upregulation of GLUT1, GLUT3, GLUT5 and GLUT12 is widely found in many types of cancers including breast, bladder, cervical, colon, colorectal, oesophageal, glioblastoma, gastric, head and neck, laryngeal, liver, lung, lymphoma, oral squamous cell, ovarian, pancreas, pancreatic islets, penile, prostate, clear renal cell, testis, thyroid, uterine cancers as well as from breast cancer to brain metastasis (Nishioka *et al.,* 1992; Brown and Wahl, 1993; Mellanen *et al.,* 1994; Zamora-Leon *et al.,* 1996; Higashi *et al.,* 1997; Younes *et al.,* 1997; Noguchi *et al.,* 1999; Pedersen *et al.,* 2001; Rogers *et al.,* 2003; Ayala

**Table 1.** The 14 glucose transporter gene expressions in various cancers

| Transporter | Tissue expression |
|---|---|
| GLUT1 | A wide range of cancers and tumours |
| GLUT2 | Liver, breast, pancreatic, colon and gastric carcinoma |
| GLUT3 | Lung, brain, breast, bladder, laryngeal, prostate, gastric, head and neck, ovarian and oral squamous carcinoma |
| GLUT4 | Colon, lymphoid, breast, thyroid, pancreatic and gastric carcinoma |
| GLUT5 | Breast, renal, colon, liver, testicular and lymphoid carcinoma |
| GLUT6 | Breast, pancreatic and endometrial carcinoma, uterine leiomyoma |
| GLUT8 | Endometrial and lymphoid carcinoma, multiple myeloma |
| GLUT9 | Liver, lung, skin, thyroid, kidney, adrenal, testicular and prostate carcinoma |
| GLUT10 | gastric carcinoma, lung adenocarcinoma |
| GLUT11 | Multiple myeloma, prostate carcinoma |
| GLUT12 | Breast, prostate, lung and colorectal carcinoma, rhabdomyosarcoma, oligodendroglioma, oligoastrocytoma, astrocytoma |
| GLUT13 | Lung adenocarcinoma |
| GLUT14 | Colon, gastric adenocarcinoma, glioblastoma |

*Note:* This table is mainly derived from Barron *et al.* (2016) and Wu *et al.* (2021), except information for GLUT10 (Schlößer *et al.,* 2017; Du *et al.,* 2020). GLUT 13 (Du *et al.,* 2020) and GLUT14 (Berlth *et al.,* 2015; Valli *et al.,* 2019; Sharpe *et al.,* 2021).

**Table 2.** AlphaFold2 structure prediction parameters

| Parameter | Value |
|---|---|
| Homooligomer | 1 |
| msa_method | mmseqs2 |
| msa_format | fas |
| pair_mode | unpaired |
| pair_cov | 50 |
| pair_qid | 20 |
| rank_by | pLDDT |
| use_turbo | True |
| max_msa | 512:1024 |
| show_images | True |
| num_models | 5 |
| use_ptm | True |
| num_ensemble | 1 |
| max_recycles | 3 |
| tol | 0 |
| num_samples | 1 |
| subsample_msa | True |
| num_relax | None |

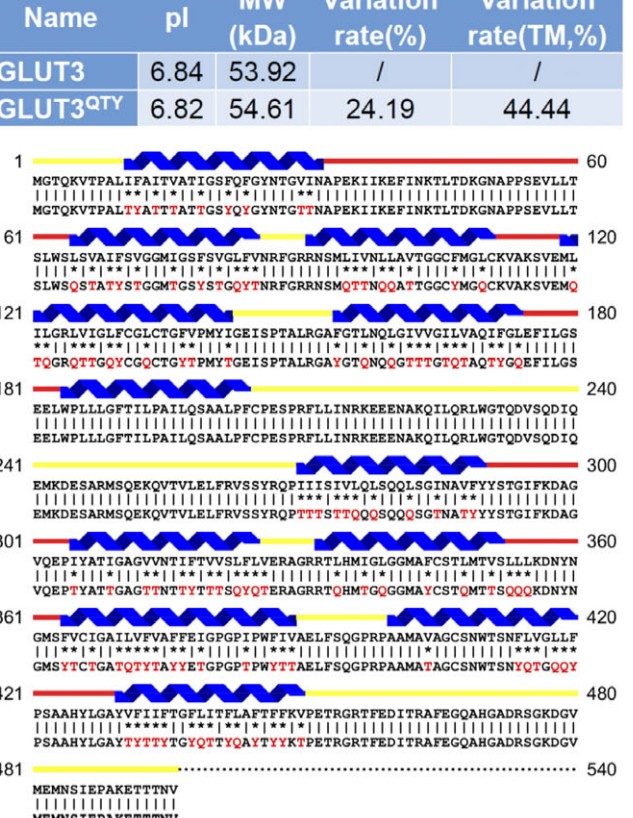

**Fig. 1.** Protein sequences alignments of native GLUT1 and GLUT3 with water-soluble QTY variants. The Q, T and Y amino acid substitutions are in red. The alpha-helical segments (blue) are shown above the protein sequences, the internal (yellow) and external (red) loops of the transporters are indicated. The symbols | and * indicate the identical and different amino acids, respectively. Characteristics of natural and QTY variants with pI, molecular weight, total variation rate and transmembrane variation rate are presented (also see Table 3). The alignment: (*a*) GLUT1 and GLUT1^QTY, (*b*) GLUT3 and GLUT3^QTY. Although there are significant overall changes, >26% for GLUT1 and >24% for GLUT3, the transmembrane (TM) domain changes, >48% for GLUT1 and >44% for GLUT3, their pI and molecular weight changes are minimal.

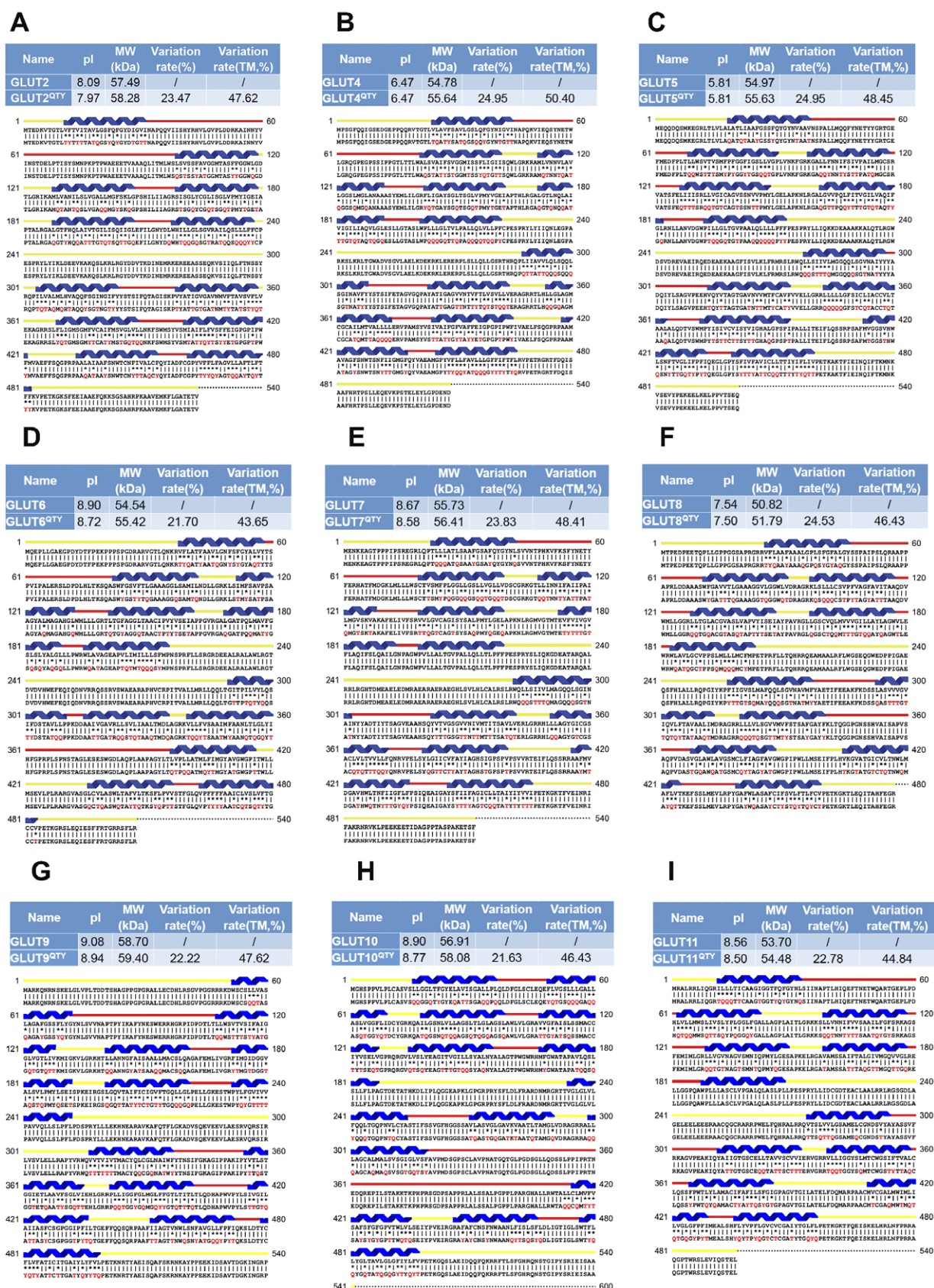

**Fig. 2.** Protein sequences alignment of 12 native glucose transporters, GLUT2 and from GLUT4 to GLUT14, with their water-soluble QTY variants. The Q, T and Y amino acid substitutions are in red. The alpha-helical segments (blue) are shown above the protein sequences, the external (red) and internal (yellow) loops of the receptors are indicated. The symbols | and * indicate the similar and different amino acids, respectively. Characteristics of natural and QTY variants with pI, molecular weight, total variation rate and membrane variation rate are presented (also see Table 1). The alignment: (*a*) GLUT2 and GLUT2^QTY, (*b*) GLUT4 and GLUT4^QTY, (*c*) GLUT5 and GLUT5^QTY, (*d*) GLUT6 and GLUT6^QTY, (*e*) GLUT7 and GLUT7^QTY, (*f*) GLUT8 and GLUT8^QTY, (*g*) GLUT9 and GLUT9^QTY, (*h*) GLUT10 and GLUT10^QTY, (*i*) GLUT11 and GLUT11^QTY, (*j*) GLUT12 and GLUT12^QTY, (*k*) GLUT13 and GLUT13^QTY and (*l*) GLUT14 and GLUT14^QTY.

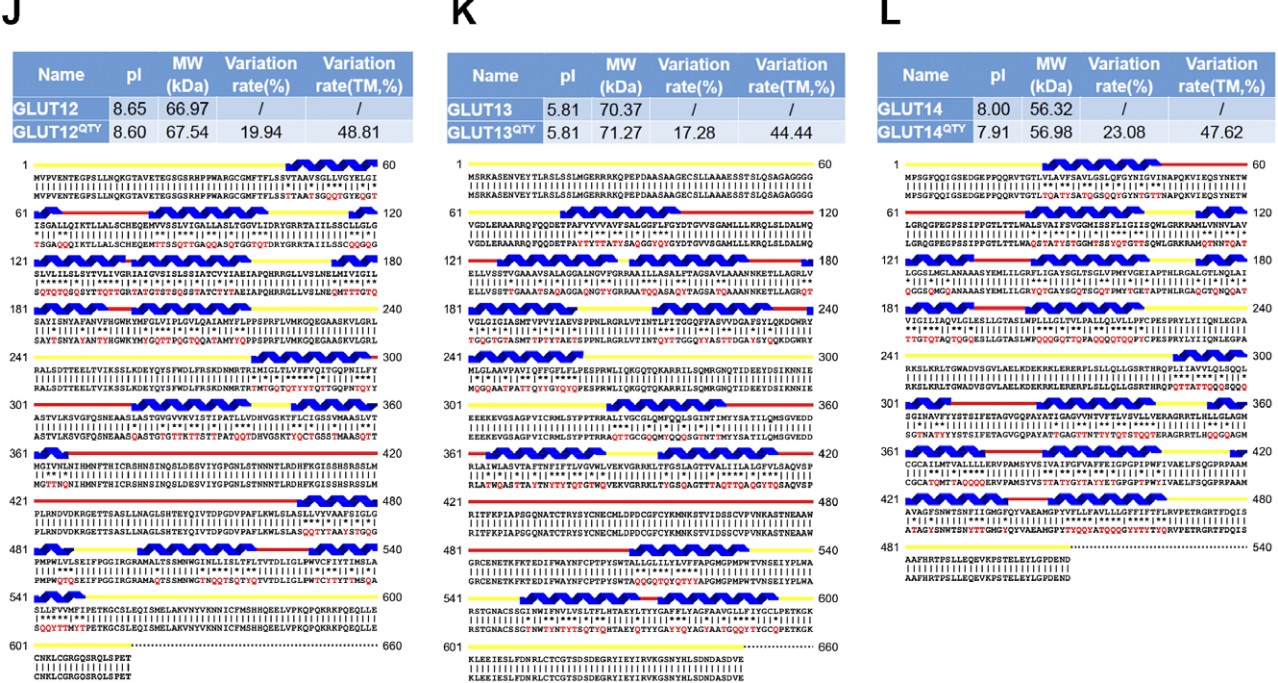

**Fig. 2.** Continued

et al., 2010; Medina Villaamil et al., 2011; Starska et al., 2015; Barron et al., 2016; Schlößer et al., 2017; White et al., 2018; Kuo et al., 2019; Heydarzadeh et al., 2020; Achalandabaso-Boira et al., 2020). Overexpression of GLUT2 is found in cancers of the breast, colon, liver, pancreas and small intestine (Tomita, 1999; Godoy et al., 2006; Hamann et al., 2018). Overexpression of GLUT4 is found in breast, gastric, and myeloma, muscle (McBrayer et al., 2012; Guo et al. 2021); overexpression of GLUT6 is found in gastric and endometrial cancer and testis (Byrne et al., 2014; Schlößer et al., 2017; Caruana and Byrne, 2020); overexpression of GLUT7 is found in benign prostate cancer (Reinicke et al., 2012); overexpression of GLUT8 is found in myeloma (McBrayer et al., 2012); overexpression of GLUT9 is found in adrenal, heart, kidney, liver, both benign and cancerous prostate (Godoy et al., 2006); overexpression of GLUT11 is found in multiple myeloma and prostate (McBrayer et al., 2012) and overexpression of GLUT14 is found in colon and glioblastoma (Valli et al., 2019; Sharpe et al., 2021). On the other hand, overexpression of GLUT10, GLUT12 and GLUT13 have also been found to be associated with better outcomes in lung adeno-carcinoma (Du et al., 2020). Furthermore, a recent study showed that GLUT12 is expressed in insulin-sensitive tissues adipose tissue and it perhaps could modulate sugar absorption in physio-logical and pathophysiological obesity (Gil-Iturbe et al., 2019).

Thus far, only the molecular structures of GLUT1, GLUT3 and GLUT4 have been elucidated (Deng et al., 2015; Custódio et al., 2021; Yuan et al., 2022). The molecular structures of other twelve GLUTs remain to be determined. Since these transporters form 12 transmembrane helices, the structural determination of these transporters requires systematic detergent screens before protein purification can be carried out. The difficulties in obtaining struc-tures of transmembrane species experimentally are well-known (Vinothkumar and Henderson, 2010).

AlphaFold2 and RoseTTAFold were introduced in July 2021 as an artificial intelligence (AI) revolutionary computational tool for the accurate prediction of protein structures (Baek et al., 2021; Jumper et al., 2021; Varadi et al., 2022). Since its introduction, both AlphaFold2 and RoseTTAFold have already made a significant impact on our understanding of the molecular structure of numer-ous proteins that were previously inaccessible. However, investiga-tors, biotech, and the pharmaceutical industry are still very interested in studying the physical structures of proteins, especially membrane transporters, since the structures are vital to under-standing how glucose and other sugars are transported across cell membranes.

We previously applied the glutamine, threonine, tyrosine (QTY) code to design several detergent-free transmembrane (TM) protein chemokine receptors and cytokine receptors for various uses using conventional computing programs (Zhang et al., 2018). The expressed proteins exhibited predicted characteristics and retained ligand-binding activity (Zhang et al., 2018; Qing et al., 2019, 2020; Tao et al., 2022; Hao et al., 2020; Tegler et al., 2020). Later we prepared QTY variant protein structure predictions using Alpha-Fold2, achieving results in hours (Skuhersky et al., 2021) rather than 4–5 weeks for each molecular simulation using GOMoDo, AMBER and YASARA programs (Zhang et al., 2018; Qing et al., 2019; Tegler et al., 2020). Here, we use AlphaFold2 to design water-soluble QTY variants of the 14 GLUTs, and to make comparison with the native structure. In addition to targeting the glucose uptake activity of cancer cells, the motivation to design these water-soluble GLUTs is to find many additional applications, such as ultrasensi-tive glucose sensing devices, as water-soluble antigens to generate valuable and specific therapeutic monoclonal antibodies to block cancer cell energy supplies. Working with water-soluble QTY variants may substantially accelerate the discovery and develop-ment of therapeutic and diagnostic biologicals.

## Materials and methods

### Protein sequence alignments and other characteristics

The native sequences for GLUT1 through GLUT14 and their QTY-variant sequences were aligned using the same methods previously described (Zhang *et al.*, 2018; Qing *et al.*, 2019). The molecular weights (MW) and pI values of the proteins were calculated using the service provided by Expasy (https://web.expasy.org/compute_pi/). For detailed transporter protein sequence information, the predicted 2D structures and hydrophobicity change, please see Supplementary Fig. 1.

### AlphaFold2 predictions

Structure predictions of the QTY variants were performed using the AlphaFold2 (Jumper *et al.*, 2021; Varadi *et al.*, 2022) software following the instructions at the website https://github.com/sokrypton/ColabFold on 2 × 20 Intel Xeon Gold 6248 cores, 384 GB RAM, and a Nvidia Volta V100 GPU. Other AlphaFold2-predicted structures were obtained from the European Bioinformatics Institute (EBI, https://alphafold.ebi.ac.uk) and are also available at Uniprot website https://www.uniprot.org. Each UniProt ID from the dataset was extended with ID, entry name, description, and FASTA sequence. The data was taken from UniProt using a custom Python code. The FASTA sequences were converted into their soluble versions using the QTY method (https://pss.sjtu.edu.cn/), followed by Protter 2D diagram plotting (http://wlab.ethz.ch/protter/start/). These steps were optimised via Python libraries for web applications such as requests and splinter.

### Superposed structures

The published X-ray crystal structures of native GLUT1 (PDB: *6THA*, 2.4 Å) and GLUT3 (PDB: *4ZW9*, 1.5 Å) were obtained from the protein data bank (PDB), https://www.rcsb.org. AlphaFold2 predictions of 12 native GLUTs and their QTY variants were carried out using the AlphaFold2 program at https://github.com/sokrypton/ColabFold. All 14 native GLUT sequences are obtained from Uniprot https://www.uniprot.org. The X-ray crystal structures and predicted structures were aligned with PyMOL.

### Structure visualisation

We used two programs for structure visualisation: PyMOL https://pymol.org/2/ and UCSF Chimera https://www.rbvi.ucsf.edu/chimera/. All superposed models were produced via PyMOL, whilst Chimera was used for hydrophobicity representation.

### Data availability of AlphaFold2 predicted water-soluble QTY variants

The AlphaFold2 predicted protein structures are at EBI (https://alphafold.ebi.ac.uk; Table 2). The QTY code designed water-soluble GLUT1–14 variants are reported in this paper and for more detail information, please go to the website: https://github.com/eva-smorodina/glucose-transporters.

## Results and discussions

### Protein sequence alignments and other characteristics

We aligned the native GLUTs with their QTY variants. Despite significant QTY replacement of hydrophobic residues overall (~17–26%), especially in the transmembrane domains (~44–50%)

in the GLUTs, the isoelectric-focusing point pI and molecular weight remain rather similar (Figs 1 and 2 and Table 3). This is because Q, T, Y amino acids do not introduce any charges, they only introduce water-soluble side chains. Q (glutamine) side chains form four water hydrogen bonds, two donors through —NH$_2$, and two acceptors through oxygen on —C=O; the sidechains —OH of T (threonine) and Y (tyrosine) form three water hydrogen bonds, one donor from H (hydrogen) and two acceptors from O (oxygen).

The QTY code selects three neutrally polar amino acids: glutamine (Q), threonine (T) and tyrosine (Y) to replace four hydrophobic amino acids leucine (L), isoleucine (I), valine (V) and phenylalanine (F), since their electron density maps share remarkable structure similarities between L *versus* Q, I,V *versus* T and F *versus* Y (Zhang *et al.*, 2018). After applying the QTY code, the hydrophobic amino acids in the transmembrane segments are replaced by Q, T and Y, therefore the transmembrane segments are no longer hydrophobic. For example, the overall proteins,

**Table 3.** Characteristics of native glucose transporters and their water-soluble QTY variants

| Name | RMSD | pI | MW (KD) | TM variation (%) | Total variation (%) |
|---|---|---|---|---|---|
| GLUT1 | — | 8.92 | 54.08 | — | — |
| GLUT1$^{QTY}$ | 1.545 Å | 8.73 | 54.80 | 48.28 | 26.83 |
| GLUT2 | — | 8.09 | 57.49 | — | — |
| GLUT2$^{QTY}$ | 3.058 Å | 7.97 | 58.28 | 47.62 | 23.47 |
| GLUT3 | — | 6.84 | 53.92 | — | — |
| GLUT3$^{QTY}$ | 1.025 Å | 6.82 | 54.61 | 44.44 | 24.19 |
| GLUT4 | — | 6.47 | 54.78 | — | — |
| GLUT4$^{QTY}$ | 0.764 Å | 6.47 | 55.64 | 50.4 | 24.95 |
| GLUT5 | — | 5.81 | 54.97 | — | — |
| GLUT5$^{QTY}$ | 0.712 Å | 5.81 | 55.63 | 48.45 | 24.95 |
| GLUT6 | — | 8.90 | 54.54 | — | — |
| GLUT6$^{QTY}$ | 1.910 Å | 8.72 | 55.42 | 43.65 | 21.70 |
| GLUT7 | — | 8.67 | 55.73 | — | — |
| GLUT7$^{QTY}$ | 0.470 Å | 8.58 | 56.41 | 48.41 | 23.83 |
| GLUT8 | — | 7.54 | 50.82 | — | — |
| GLUT8$^{QTY}$ | 1.169 Å | 7.50 | 51.79 | 46.43 | 24.53 |
| GLUT9 | — | 9.08 | 58.70 | — | — |
| GLUT9$^{QTY}$ | 0.593 Å | 8.94 | 59.40 | 47.62 | 22.22 |
| GLUT10 | — | 8.90 | 56.91 | — | — |
| GLUT10$^{QTY}$ | 1.186 Å | 8.77 | 58.08 | 46.43 | 21.63 |
| GLUT11 | — | 8.56 | 53.70 | — | — |
| GLUT11$^{QTY}$ | 1.502 Å | 8.50 | 54.48 | 44.84 | 22.78 |
| GLUT12 | — | 8.65 | 66.97 | — | — |
| GLUT12$^{QTY}$ | 3.590 Å | 8.60 | 67.54 | 48.81 | 19.94 |
| GLUT13 | — | 5.81 | 70.37 | — | — |
| GLUT13$^{QTY}$ | 0.881 Å | 5.81 | 71.27 | 44.44 | 17.28 |
| GLUT14 | — | 8.00 | 56.32 | — | — |
| GLUT14$^{QTY}$ | 2.720 Å | 7.91 | 56.98 | 47.62 | 17.28 |

Abbreviations: MW, molecular weight; pI, isoelectric focusing; RMSD, residue mean-square distance in Å; TM, transmembrane; —, not applicable.

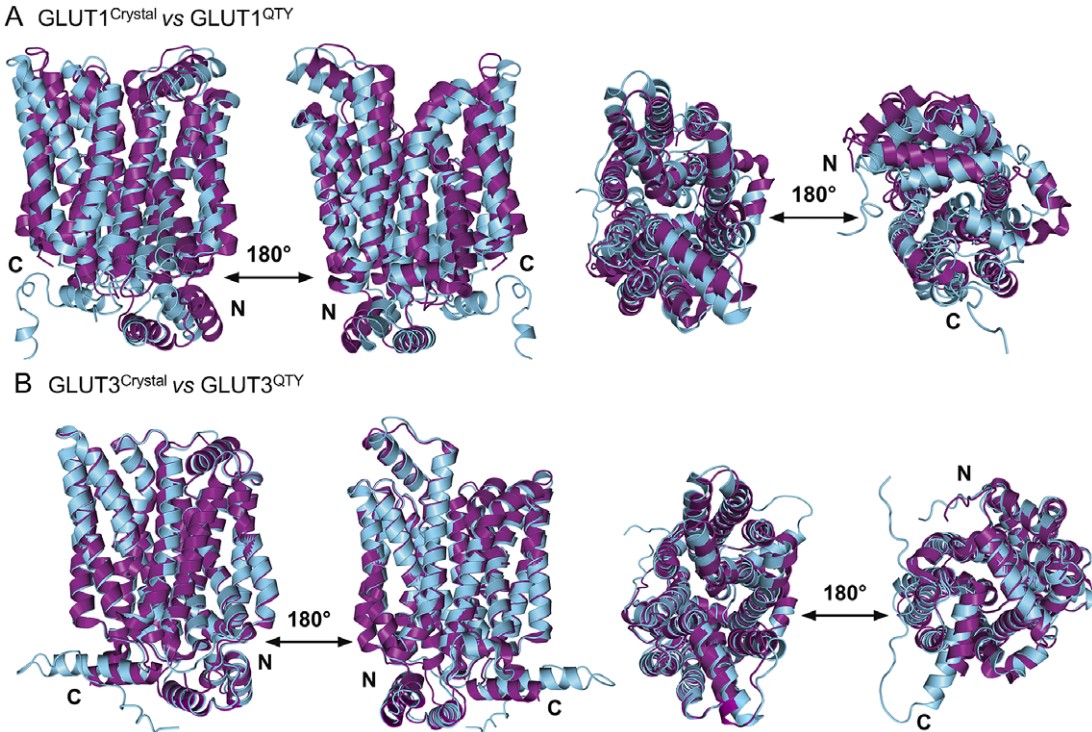

**Fig. 3.** Superposed two transporter crystal structures of GLUT1$^{Crystal}$ and GLUT3$^{Crystal}$ with AlphaFold2 predicted QTY water-soluble variants GLUT1$^{QTY}$ and GLUT3$^{QTY}$. For each superposition, four structures are shown, front (left), back (middle left) and view from top axis (middle right) and bottom (right). The X-ray crystal structures of native GLUT1 (6THA, 2.4 Å, P11166), GLUT3 (4ZW9, 1.5 Å, P11169), are obtained from the protein data bank (PDB). *N*- and *C*-termini are labelled. (*a*) The crystal structure GLUT1$^{Crystal}$ (magenta) is superposed with AlphaFold2 predicted water-soluble variant GLUT1$^{QTY}$ (cyan). The RMSD is 1.545 Å for GLUT1 and GLUT1$^{QTY}$. Following the same order, the superposed crystal structure and AlphaFold2 predicted structures of (*b*) GLUT3$^{Crystal}$ (magenta), water-soluble variant GLUT3$^{QTY}$ (cyan). The RMSD is 1.025 Å for GLUT3$^{Crystal}$ and GLUT3$^{QTY}$. These results show that the glucose transporter water-soluble QTY variants share remarkable structural similarity despite >44% QTY replacements in transmembrane alpha-helices.

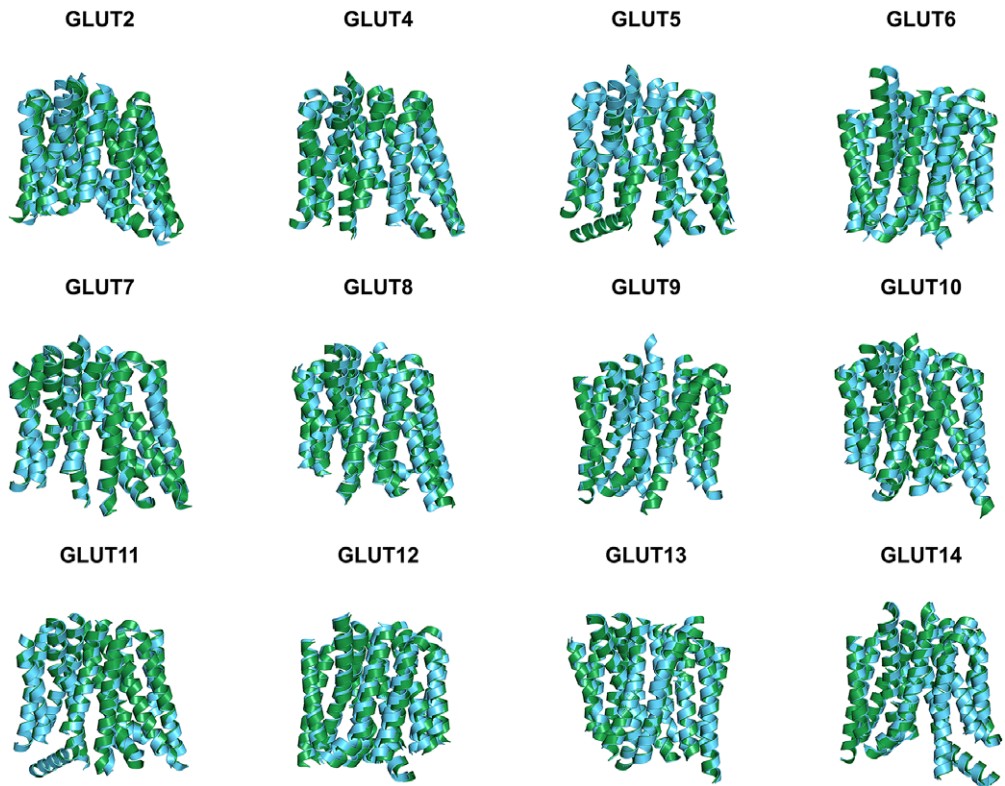

**Fig. 4.** Superposed AlphaFold2 predicted 12 native glucose transporters GLUT2, and from GLUT4 to GLUT14 and their QTY water-soluble variants. For clarity, large *N*- and *C*-termini are removed. The predicted native structures (green) and their water-soluble QTY variants (cyan). The RMSD in Å for the superposed structures are in (). (*a*) GLUT2 and GLUT2$^{QTY}$ (3.058 Å) (*b*) GLUT4 and GLUT4$^{QTY}$ (0.764 Å), (*c*) GLUT5 and GLUT5$^{QTY}$ (0.712 Å), (*d*) GLUT6 and GLUT6$^{QTY}$ (1.910 Å), (*e*) GLUT7 and GLUT7$^{QTY}$ (0.470 Å), (*f*) GLUT8 and GLUT8$^{QTY}$ (1.169 Å), (*g*) GLUT9 and GLUT9$^{QTY}$ (0.593 Å), (*h*) GLUT10 and GLUT10$^{QTY}$ (1.186 Å), (*i*) GLUT11 and GLUT11$^{QTY}$ (1.502 Å), (*j*) GLUT12 and GLUT12$^{QTY}$ (3.590 Å), (*k*) GLUT13 and GLUT13$^{QTY}$ (0.881 Å) and (*l*) GLUT14 and GLUT14$^{QTY}$ (2.720 Å). Please also see Table 3.

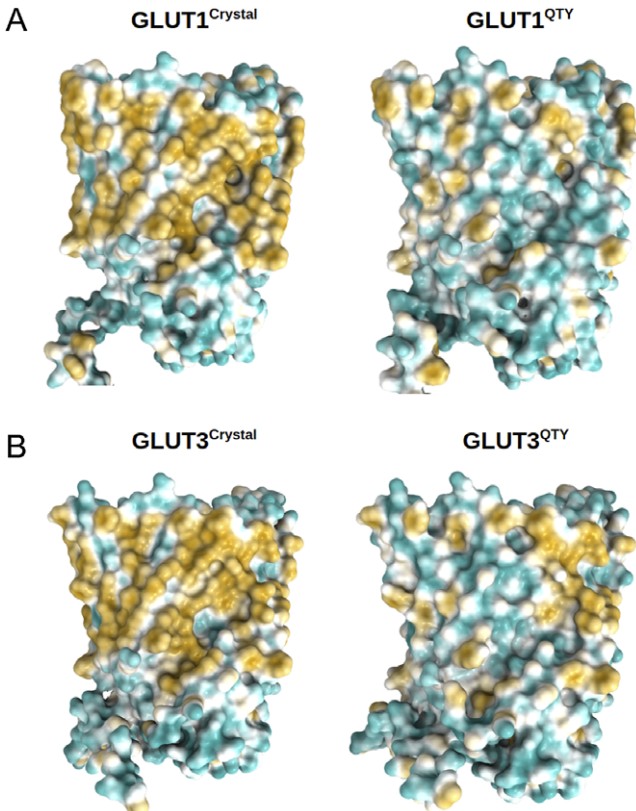

**Fig. 5.** Surface hydrophobic patch of crystal structures of native glucose transporters GLUT1 and GLUT3 and AlphaFold2 predicted water-soluble QTY variants. The native glucose transporters mostly expose hydrophobic residues leucine (L), isoleucine (I), valine (V) and phenylalanine (F) facing outside to the hydrophobic lipid bilayer in cell membrane. After replacing the L, I, V, F with polar amino acids, glutamine (Q), threonine (T) and tyrosine (Y), the surfaces are much less hydrophobic. The large surface hydrophobic patch (yellow colour) of the native receptors from X-ray crystal structures: (*a*) GLUT1$^{Crystal}$ and GLUT1$^{QTY}$; (*b*) GLUT3$^{Crystal}$ and GLUT3$^{QTY}$. The hydrophobic patch is significantly reduced on the transmembrane domains for the water-soluble QTY variants. These QTY variants become water-soluble without any detergent. The large *N*- and *C*-termini are removed for clarity of direct comparisons.

GLUT1 differs 26.83% from its QTY variant and GLUT13 differs 17.28% from its QTY variant. Most differences are >20% (Figs 1 and 2 and Table 3).

The isoelectric-focusing points (pI) vary, some in the acidic and some in the basic range. For example, native GLUT1, GLUT2, GLUT6, GLUT7, GLUT9, GLUT10, GLUT11, GLUT12 and GLUT14 have basic pIs > 8.0. On the other hand, GLUT5 and GLUT13 have acidic pIs < 6.0. Others including GLUT3, GLUT4, GLUT8 have near neutral pIs, ~ 6.84–7.5 (Table 3). Notably, the isoelectrical focusing points (pIs) are identical for the native and QTY variants for GLUT4, GLUT5 and GLUT13 despite significant QTY sequence replacement. The pIs in the QTY variants have little changes, and in some examples, there are no change. This is because that three amino acids Q, T, Y do not bear any charges at neutral pH. Thus, introductions of these amino acids Q, T, Y do not significantly alter the pI. This is important since altered pI may result the non-specific interactions.

Furthermore, although there are between ~43.6 and ~50% transmembrane QTY replacements, the molecular weights of the native and QTY variants differ by only a few hundreds of Daltons.

This is because the addition of −OH groups from QTY amino acids increased their molecular weights (Figs 1 and 2 and Table 3).

### Superposition of native transporters and their water-soluble QTY variants

In this study, we compare the molecular structures of native GLUTs and their QTY variants. Since the crystal structures of native receptors GLUT1 (PDB: *6THA*) (Deng *et al.,* 2015) and GLUT3 (PDB: *4ZW9*) (Custódio *et al.,* 2021) are already available, the superposed structures were carried out for: GLUT1 *versus* GLUT1$^{QTY}$, and GLUT3 *versus* GLUT3$^{QTY}$. Therefore, their crystal structures and the AlphaFold2 predicted QTY variant structures can be directly compared.

The native structures and water-soluble QTY variants superposed very well, the RMSD is GLUT1 and GLUT1$^{QTY}$ is 1.545 Å and for GLUT3 and GLUT3$^{QTY}$ is 1.025 Å (Fig. 3, Table 3 and Supplementary Table 1). In the first sets of structures for GLUT1 and GLUT3, they are superposed among: (i) the X-ray crystal determined structures (magenta) GLUT1 (*6THA*) and GLUT3 (*4ZW9*), and the (ii) AlphaFold2 predicted water-soluble QTY variants (cyan). As seen from Fig. 3, these structures clearly superposed well, the structures are viewed from front, back, top and bottom. The results suggest that these structures share very similar folds despite >44% QTY amino acid replacement in the transmembrane helices in the water-soluble QTY variants. These closely superposed structures perhaps confirm that the AlphaFold2's predictions are highly accurate, since the predicted native structures are directly superposable with the experimentally determined X-ray crystal structures. These results also suggest the native GLUT and their water-soluble QTY variants share remarkable structural similarity.

Since the molecular structures for the other 12 native glucose transporters (GLUT2, and GLUT4 to GLUT14) are not yet available, we used AlphaFold2 predictions. The residue mean-square distances (RMSD in Å) for the superposed structures are also shown in Fig. 4 and Table 3. The pairwise examples are: GLUT1 *versus* GLUT1$^{QTY}$ (1.545 Å), GLUT2 *versus* GLUT2$^{QTY}$ (3.058 Å), GLUT3 *versus* GLUT3$^{QTY}$ (1.025 Å), GLUT4 *versus* GLUT4$^{QTY}$ (0.764 Å), GLUT5 *versus* GLUT5$^{QTY}$ (0.712 Å), GLUT6 *versus* GLUT6$^{QTY}$ (1.910 Å), GLUT7 *versus* GLUT7$^{QTY}$ (0.470 Å), GLUT8 *versus* GLUT8$^{QTY}$ (1.169 Å), GLUT9 *versus* GLUT9$^{QTY}$ (0.593 Å), GLUT10 *versus* GLUT10$^{QTY}$ (1.186 Å), GLUT11 *versus* GLUT11$^{QTY}$ (1.502 Å), GLUT12 *versus* GLUT12$^{QTY}$ (3.590 Å), GLUT13 *versus* GLUT13$^{QTY}$ (0.881 Å) and GLUT14 *versus* GLUT14$^{QTY}$ (2.720 Å) (Table 3 and Supplementary Table 2). These AlphaFold2-predicted native and their water-soluble GLUT structures superposed very well suggesting that they share similar molecular structures despite the large transmembrane variations (44–50%).

### Analysis of the hydrophobic surface of native transporters and the water-soluble QTY variants

The native GLUTs are highly hydrophobic, especially in the 12TM helical domains, thus they are intrinsically water-insoluble and require detergents to solubilise them after removing them from the lipid bilayer membranes. Without the appropriate detergents, they immediately aggregate, precipitate and lose their biological functions.

The 12TM domains are directly embedded in the hydrophobic lipid bilayer so the hydrophobic side chains of amino acids leucine

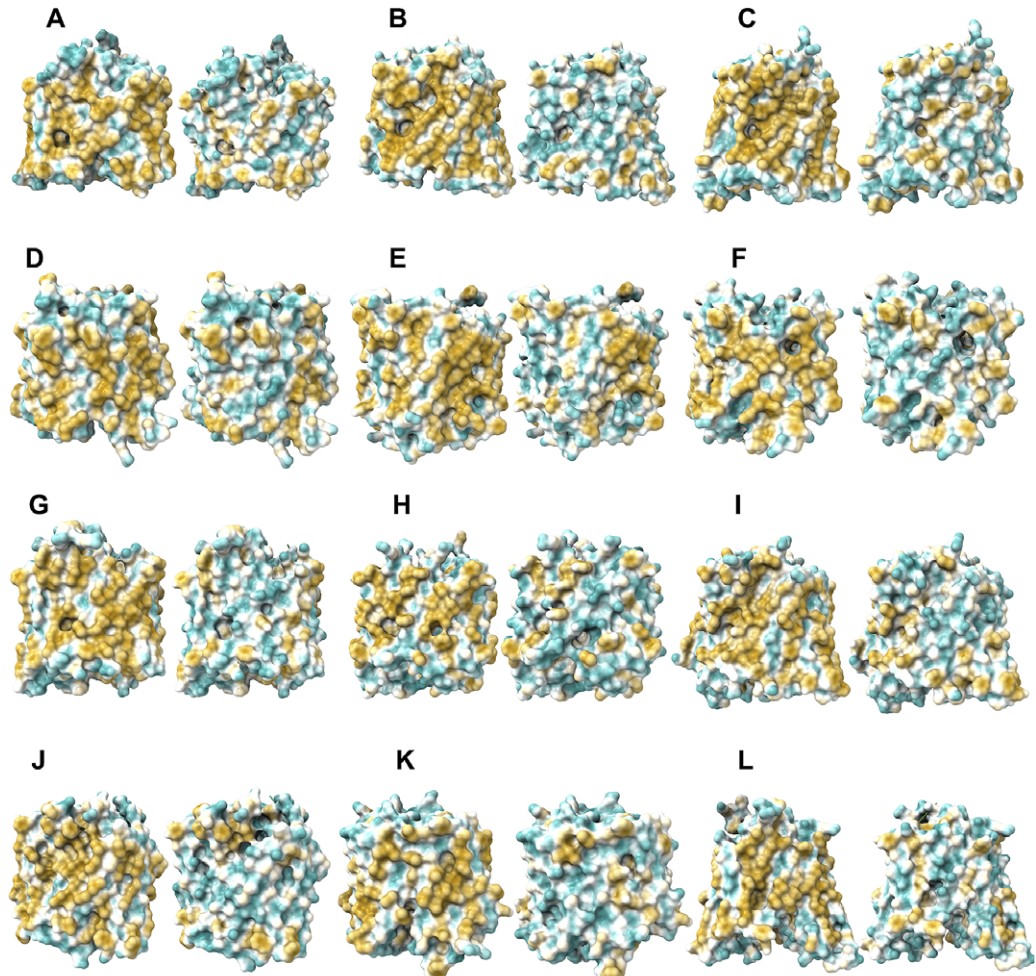

**Fig. 6.** Surface hydrophobic patch of AlphaFold2 predicted structures of native glucose transporters GLUT2, and from GLUT4 to GLUT14 and their water-soluble QTY variants. The pairwise of AlphaFold2 predicted native structures with large surface hydrophobic patch (yellowish colour), and the water-soluble QTY variant transporters (cyan colour): (*a*) GLUT2 and GLUT2$^{QTY}$, (*b*) GLUT4 and GLUT4$^{QTY}$, (*c*) GLUT5 and GLUT5$^{QTY}$, (*d*) GLUT6 and GLUT6$^{QTY}$, (*e*) GLUT7 and GLUT7$^{QTY}$, (*f*) GLUT8 and GLUT8$^{QTY}$, (*g*) GLUT9 and GLUT9$^{QTY}$, (*h*) GLUT10 and GLUT10$^{QTY}$, (*i*) GLUT11 and GLUT11$^{QTY}$, (*j*) GLUT12 and GLUT12$^{QTY}$, (*k*) GLUT13 and GLUT13$^{QTY}$ and (*l*) GLUT14 and GLUT14$^{QTY}$. The *N*- and *C*-termini are removed for clarity of direct comparisons.

(L), isoleucine (I), valine (V) and phenylalanine (F) directly interact with the lipid molecules excluding water. Thus, they display highly hydrophobic patches on the 12TM domain (Figs 5 and 6 and Supplementary Fig. 1).

On the other hand, after systematic QTY replacement of hydrophobic amino acids L, I, V, F, with hydrophilic amino acids Q, T, Y, these hydrophobic patches become largely reduced (Figs 5 and 6). The QTY transformation from hydrophobic 12TM to hydrophilic 12TM did not significantly alter the alpha-helix structures. This would have been rather unexpected prior to obtaining the experimental results reported in recent publications (Zhang *et al.,* 2018; Qing *et al.,* 2019; Hao *et al.,* 2020). However, that work demonstrated that QTY-designed chemokine receptors and cytokine receptors retained structure stability, integrity and ligand-binding function (Zhang *et al.,* 2018; Qing *et al.,* 2019, 2020; Hao *et al.,* 2020; Tegler *et al.,* 2020).

Nature has already evolved three distinct types of alpha-helices: (1) the hydrophilic alpha-helix such as found in haemoglobin and many other water-soluble enzymes and circulating proteins such as growth factors, cytokines and antibodies; (2) the hydrophobic alpha-helix such as in transporters and other integral

transmembrane proteins found in G protein-coupled receptors, ion channels, photosynthesis systems and (3) amphiphilic alpha-helices with both hydrophilic and hydrophobic amino acid residues. These three types of alpha-helices have nearly identical molecular structures, irrespective of the hydrophobicity and hydrophilicity (Pauling and Corey, 1951; Branden and Tooze, 1999; Fersht, 2017). This insight is the structural basis of the QTY code.

### *AlphaFold2 predictions*

For over six decades, structural biologists and protein scientists have sought to predict how proteins fold naturally and rapidly. We can now study protein structure in more detail *in silico*, and obtain previously unattainable protein structures, especially integral transmembrane proteins, through AlphaFold2 predictions, at least in the framework.

It is estimated that ~20–30% genes code for membrane proteins in most organisms (Krogh *et al.,* 2001) and ~25.86% of genes (5,139 among 21,416 annotated genes) in the human genome code for membrane proteins (Fagerberg *et al.,* 2010). However, determining the structure of even a single transmembrane protein is a daunting

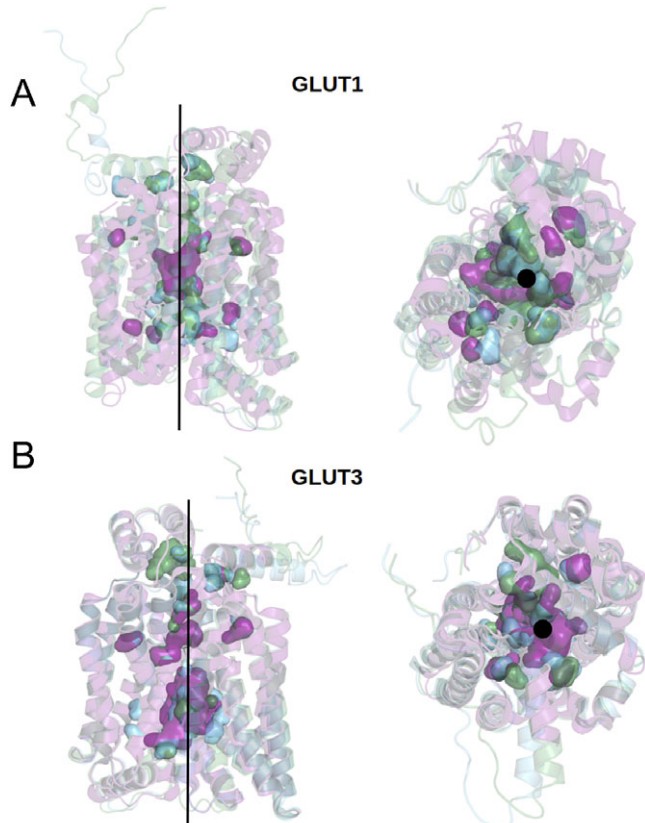

**Fig. 7.** Superposed structures of crystal, AlphaFold2 predicted native and QTY variants with the glucose channel. (*a*) GLUT1$^{Crystal}$, GLUT1$^{Native}$ and GLUT1$^{QTY}$, (*b*) GLUT3$^{Crystal}$, GLUT3$^{Native}$ and GLUT3$^{QTY}$. Black lines and dots and surface representation show the position of the glucose channel in three structures: crystal structures (magenta), AlphaFold2 predicted native (green) and AlphaFold2 predicted water-soluble QTY variants (cyan).

task, with many hurdles along the way, from gene expression, protein production, detergent selection, purification, to maintaining the stability, integrity and functionality to avoid aggregation. The determination of integral transmembrane protein structures lags far behind of those of water-soluble proteins.

We can now approach transmembrane structures computationally, using AlphaFold2 predictions, comparing the native structure with a proposed water-soluble QTY variant, and then express the water-soluble structure *in vitro*. We also modelled the glucose in its transporter channel (Fig. 7).

DeepMind AlphaFold's team has already deposited ~ 1 million open protein structures in the EBI (https://alphafold.ebi.ac.uk). The number will continuously increase over time.

We previously used the AlphaFold2 accurate prediction tool to predict water-soluble chemokine receptors and compare them to the known experimentally determined structures (Skuhersky *et al.,* 2021). The speed and accuracy of AlphaFold2 predictions of our designed receptors is unprecedented. Instead of taking weeks or days to predict one structure, AlphaFold2 can compute a new structure in hours, or even minutes for smaller proteins. AlphaFold2 significantly accelerates studies of protein structures, stabilities, the design of new proteins, the discovery of new protein interactions, and perhaps new functions that were previously unknown through expensive and time-consuming experimental studies.

### The location of glucose in the GLUT1$^{native}$ and GLUT1$^{QTY}$ variant

The crystal structure of native GLUT1 was determined with a glucose in the transporter channel (Deng *et al.,* 2015). It is interesting to know where the glucose is in the AlphaFold2 predicted water-soluble GLUT1$^{QTY}$ variant. As can be seen from superposed structures of native GLUT1 and GLUT1$^{QTY}$ variant (Fig. 7), the glucose still is in the same channel location as the native GLUT1$^{native}$. Since both native and QTY variant superposed very well, the glucose in the channel is likely to be in the same location since glucose is highly hydrophilic and the QTY variant is unlikely to reduce its interaction with other hydrophilic amino acid residues.

### Conclusion

Our study provides insight into the subtle differences between the hydrophobic and hydrophilic alpha-helices through systematically comparing experimentally determined structures GLUT1, GLUT3 and GLUT4 with AlphaFold2-predicted water-soluble QTY variant transporters, as well as AlphaFold2 predicted 11 native GLUT2 and from GLUT5 to GLUT14 and their QTY variants. Our study demonstrates the use of QTY-variant structures as a viable approach to modelling integral membrane proteins and other aggregated proteins. These GLUT water-soluble QTY variants may also be very useful for designing glucose sensing device and for drug discoveries.

**Supplementary Materials.** To view supplementary material for this article, please visit http://doi.org/10.1017/qrd.2022.6.

**Acknowledgements.** We thank Dr. Arthur Zalevsky for helpful discussions. We also thank Dorrie Langsley for English editing.

**Author contributions.** Conceptualization: S.Z.; Data curation: E.S.; Formal analysis: F.T.; Funding acquisition: D.J.; Investigation: F.T., R.Q., S.Y.; Methodology: E.S., F.T.; Project administration: S.Z.; Resources: D.J.; Software: AlphaFold2, E.S.; Supervision: S.Z.; Validation: E.S.; Visualisation: E.S.; Writing – original draft preparation: S.Z.; Writing – review and editing: E.S., F.T., R.Q., D.J., S.Y., S.Z. All authors have read and agreed to the published version of the manuscript.

**Data availability statement.** The AlphaFold2 predicted protein structures are at European Bioinformatics Institute (EBI, https://alphafold.ebi.ac.uk). The QTY code designed water-soluble variants are published in this article and at https://github.com/eva-smorodina/glucose-transporters.

**Financial support.** This work was primarily funded by Avalon GloboCare Corp.

**Conflict of interest.** Massachusetts Institute of Technology (MIT) filed several patent applications for the QTY code for GPCRs and OH$_2$ Laboratories licenced the technology from MIT to work on water-soluble GPCR variants. However, this article does not study GPCRs. S.Z. is an inventor of the QTY code and has a minor equity of OH$_2$ Laboratories shares. The funder Avalon Globo-Care Corp had no role in the design of the study; in the collection, analyses, or interpretation of data; in the writing of the manuscript, or in the decision to publish the results.

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
