## [Reviewer Report]

*Comments to Author*: This is an interesting paper reporting analysis of mutated in silico hydrophobic sequences into hydrophilic ones in transmembrane helical segments using the QTY tool in analysis of glucose transporters that are implicated in various cancers. The authors have used the AlphaFold2 predictions for potential water-soluble QTY variants of these membrane transporters by mutating the transmembrane helices making them hydrophilic. The predicted structures of mutated glucose transporters GLUT2, and GLUT4 to GLUT14 were compared with known structures of GLUT1 (PDB: 6THA) and GLUT3 (PDB: 4ZW9) revealing remarkable similarity.

This is a quite important approach that has potential vital impact on further studies glucose transporters saving significant amount of time and finances in analysis of function of this group of proteins.

It would be important ifthe authors will address some questions:

It was not clear how the solubility of the potential hydrophilic glucose transporters will change their functions. It would be good to model their interactions with glucose. It was not clear if the cancer related overexpression/upregulation glucose transporters were related to their incorporation into membranes or changes in their transport of glucose. Highly possible that this info has been published in some papers and that must be verified and analysed in silico.Plausibly that these channels by which glucose is transported though the membrane were changed, but this issue was not analysed.

What is importance of the isoelectric focusing point of glucose transporters?Is it different in the cancer cells? Hope this is known from previous publications.

Additional comments

It would be much easier for a reader to have a table where GLUT 1-14 would be listed and in this table related cancers, availability of a structure, and references would be indicated.This part of introduction is rather difficult to read and recapitulate the most important facts.It was not clear if these glucose transporters were mutated in cancerous cell and were not able to be incorporated into membranes or just did not function. That would be good to know.

It would be good to see in the figure where is the glucose channel located within the proteins obtained by X-ray.It would be important to see the same areas in silico mutants.

It would be recommended to replace throughout the entire MS the word "superimpose" to "superpose"

Superposed - Place (something) on or above something else, esp. so that they coincide. Used mostly in scientific or mathematical contexts. To place one geometric figure on top of another in such a way that all common parts coincide.

Superimpose-To place an object over another object, usually in such a way that both will be visible. Typically related to 2D images, one does not worry if they are not coinciding.

Figure 3. It would be good if the authors will check the orientations GLUT1 and GLUT3 in this figure. It seems that views (front and side) are swapped for GLUT3 compared to the view of GLU1.The last panel of GLUT3 (Fig3b) seems to be rotated in plane and in a different orientation with respect to GLUT1 as well. It is strongly recommended to indicate where are the N - and C-termini and the angles between different views. Nothing is shown in green. Please check the legend.

---

## [Reviewer Report]

*Comments to Author*: The authors have demonstrated that 14 glucose transporters’ structures and their water-soluble counterparts were solved using AlphaFold2 and their QTY tool. They have demonstrated that the native and water-soluble variants are structurally similar and the water-soluble variants are indeed hydrophilic from the decreased hydrophobic patches. I would like to request the following revisions:

1. The abbreviation QTY is not defined anywhere in the abstract or in the introduction. Please state and combine these later sentences as soon as the "QTY tool" is mentioned: "In the native structures of the transmembrane helices, there are hydrophobic amino acids leucine (L), isoleucine (I), valine (V) and phenylalanine (F). Applying the QTY code, these hydrophobic amino acids are systematically replaced by hydrophilic amino acids glutamine (Q), threonine (T) and tyrosine (Y) rendering them water- soluble."

2. The introduction should also mention RosettaFold - AlphaFold2 is not the only available tool that predicts structure from sequence. Was there a reason why AlphaFold2 was used over RosettaFold?

3. Which "conventional computing programs" were used for the previous QTY application?

4. There should be relevant references for the sentence "The difficulties in obtaining structures of transmembrane species experimentally are well-known" and for the sentence "In addition to targeting the glucose uptake activity of cancer cells, these water-soluble glucose transporters can prospectively find many additional applications, such as ultrasensitive glucose sensing devices, as water-soluble antigens to generate therapeutic monoclonal antibodies."

5. Please state the implications of the pIs changing. It’s also unclear whether the stated pIs in the sentence are for native or water-soluble variants.

6. Were the pLDDT scores high for these structures? How confident should we be in using these structures?

7. The time it takes in predicting the structure is stated in minutes-weeks but when using what kind of computing resources? Please clarify.

8. Please state the advantages and disadvantages for using the QTY method and anything that users should be aware of when applying this for other systems and using outputs from this method.

9. The SI should include all of the structures of the native and water-soluble variants of these glucose transporters (in PDB format) for researchers who can use them for MD simulations. Also, at what pH, ionic concentration, and temperature are these structures expected to be in? That would be useful information for researchers setting up the simulations.

---

## [Reviewer Report]

*Comments to Author*: Reviewer #2: The authors have demonstrated that 14 glucose transporters’ structures and their water-soluble counterparts were solved using AlphaFold2 and their QTY tool. They have demonstrated that the native and water-soluble variants are structurally similar and the water-soluble variants are indeed hydrophilic from the decreased hydrophobic patches. I would like to request the following revisions:

1. The abbreviation QTY is not defined anywhere in the abstract or in the introduction. Please state and combine these later sentences as soon as the "QTY tool" is mentioned: "In the native structures of the transmembrane helices, there are hydrophobic amino acids leucine (L), isoleucine (I), valine (V) and phenylalanine (F). Applying the QTY code, these hydrophobic amino acids are systematically replaced by hydrophilic amino acids glutamine (Q), threonine (T) and tyrosine (Y) rendering them water- soluble."

2. The introduction should also mention RosettaFold - AlphaFold2 is not the only available tool that predicts structure from sequence. Was there a reason why AlphaFold2 was used over RosettaFold?

3. Which "conventional computing programs" were used for the previous QTY application?

4. There should be relevant references for the sentence "The difficulties in obtaining structures of transmembrane species experimentally are well-known" and for the sentence "In addition to targeting the glucose uptake activity of cancer cells, these water-soluble glucose transporters can prospectively find many additional applications, such as ultrasensitive glucose sensing devices, as water-soluble antigens to generate therapeutic monoclonal antibodies."

5. Please state the implications of the pIs changing. It’s also unclear whether the stated pIs in the sentence are for native or water-soluble variants.

6. Were the pLDDT scores high for these structures? How confident should we be in using these structures?

7. The time it takes in predicting the structure is stated in minutes-weeks but when using what kind of computing resources? Please clarify.

8. Please state the advantages and disadvantages for using the QTY method and anything that users should be aware of when applying this for other systems and using outputs from this method.

9. The SI should include all of the structures of the native and water-soluble variants of these glucose transporters (in PDB format) for researchers who can use them for MD simulations. Also, at what pH, ionic concentration, and temperature are these structures expected to be in? That would be useful information for researchers setting up the simulations.

Reviewer #3: This is an interesting paper reporting analysis of mutated in silico hydrophobic sequences into hydrophilic ones in transmembrane helical segments using the QTY tool in analysis of glucose transporters that are implicated in various cancers. The authors have used the AlphaFold2 predictions for potential water-soluble QTY variants of these membrane transporters by mutating the transmembrane helices making them hydrophilic. The predicted structures of mutated glucose transporters GLUT2, and GLUT4 to GLUT14 were compared with known structures of GLUT1 (PDB: 6THA) and GLUT3 (PDB: 4ZW9) revealing remarkable similarity.

This is a quite important approach that has potential vital impact on further studies glucose transporters saving significant amount of time and finances in analysis of function of this group of proteins.

It would be important ifthe authors will address some questions:

It was not clear how the solubility of the potential hydrophilic glucose transporters will change their functions. It would be good to model their interactions with glucose. It was not clear if the cancer related overexpression/upregulation glucose transporters were related to their incorporation into membranes or changes in their transport of glucose. Highly possible that this info has been published in some papers and that must be verified and analysed in silico.Plausibly that these channels by which glucose is transported though the membrane were changed, but this issue was not analysed.

What is importance of the isoelectric focusing point of glucose transporters?Is it different in the cancer cells? Hope this is known from previous publications.

Additional comments

It would be much easier for a reader to have a table where GLUT 1-14 would be listed and in this table related cancers, availability of a structure, and references would be indicated.This part of introduction is rather difficult to read and recapitulate the most important facts.It was not clear if these glucose transporters were mutated in cancerous cell and were not able to be incorporated into membranes or just did not function. That would be good to know.

It would be good to see in the figure where is the glucose channel located within the proteins obtained by X-ray.It would be important to see the same areas in silico mutants.

It would be recommended to replace throughout the entire MS the word "superimpose" to "superpose"

Superposed - Place (something) on or above something else, esp. so that they coincide. Used mostly in scientific or mathematical contexts. To place one geometric figure on top of another in such a way that all common parts coincide.

Superimpose-To place an object over another object, usually in such a way that both will be visible. Typically related to 2D images, one does not worry if they are not coinciding.

Figure 3. It would be good if the authors will check the orientations GLUT1 and GLUT3 in this figure. It seems that views (front and side) are swapped for GLUT3 compared to the view of GLU1.The last panel of GLUT3 (Fig3b) seems to be rotated in plane and in a different orientation with respect to GLUT1 as well. It is strongly recommended to indicate where are the N - and C-termini and the angles between different views. Nothing is shown in green. Please check the legend.